# Neurotrophins and Other Growth Factors in the Pathogenesis of Alzheimer’s Disease

**DOI:** 10.3390/life13030647

**Published:** 2023-02-26

**Authors:** Tadahiro Numakawa, Ryutaro Kajihara

**Affiliations:** 1Department of Cell Modulation, Institute of Molecular Embryology and Genetics, Kumamoto University, Kumamoto 860-0811, Japan; 2Department of Biomedical Laboratory Sciences, Faculty of Life Science, Kumamoto University, Kumamoto 862-0976, Japan

**Keywords:** BDNF, TrkB, p75NTR, Alzheimer’s disease, GDNF, IGF-I, bFGF, flavonoids, signaling

## Abstract

The involvement of the changed expression/function of neurotrophic factors in the pathogenesis of neurodegenerative diseases, including Alzheimer’s disease (AD), has been suggested. AD is one of the age-related dementias, and is characterized by cognitive impairment with decreased memory function. Developing evidence demonstrates that decreased cell survival, synaptic dysfunction, and reduced neurogenesis are involved in the pathogenesis of AD. On the other hand, it is well known that neurotrophic factors, especially brain-derived neurotrophic factor (BDNF) and its high-affinity receptor TrkB, have multiple roles in the central nervous system (CNS), including neuronal maintenance, synaptic plasticity, and neurogenesis, which are closely linked to learning and memory function. Thus, many investigations regarding therapeutic approaches to AD, and/or the screening of novel drug candidates for its treatment, focus on upregulation of the BDNF/TrkB system. Furthermore, current studies also demonstrate that GDNF, IGF1, and bFGF, which play roles in neuroprotection, are associated with AD. In this review, we introduce data demonstrating close relationships between the pathogenesis of AD, neurotrophic factors, and drug candidates, including natural compounds that upregulate the BDNF-mediated neurotrophic system.

## 1. Introduction

The neurotrophin family, which includes nerve growth factor (NGF), BDNF, neurotrophin-4/5 (NT-4/5), and NT-3, has a variety of neuronal roles in the peripheral nervous system (PNS) and CNS. In particular, the expression of BDNF and its receptor TrkB is abundant in the brain, and BDNF/TrkB-mediated intracellular signaling (which includes mitogen-activated protein kinase/extracellular signal-regulated kinase (MAPK/ERK), phosphoinositide 3-kinase (PI3K)/Akt, and the phospholipase (PLC) γ pathways) contributes to neuronal cell survival and synaptic plasticity [1]. While these neurotrophins are initially synthesized as proneurotrophins (such as proNGF and proBDNF), which are larger precursors and associate with p75NTR for cell death induction, mature neurotrophins (NGF, BDNF, etc.), produced through the proteolytic cleavage process, bind to Trk receptors, leading to neuroprotection and the positive regulation of synaptic function. As expected, a growing number of studies suggest that decreased BDNF is involved in the aging and age-related cognitive dysfunction observed in AD. In contrast, upregulation of the BDNF/TrkB system, stimulated by drug candidates, has been intensively investigated to achieve effective treatment for AD.

Although a growing number of people are suffering from different types of dementia, AD is the most common cause of dementia, and is one of the top global healthcare problems [2]. As possible molecular mechanisms, it has been demonstrated that the aggregation of both amyloid-beta (Aβ) and the tau protein tangle formation contribute to the neuronal vulnerability observed in AD [2]. Recently, AD has been considered as a disease caused by various factors, including increasing age, head injuries, infections and environmental factors, in addition to cholinergic neuronal dysfunction and Aβ aggregation [3]. Interestingly, recent studies suggest that depression is considered to be a risk factor for AD [4]. There is growing evidence that demonstrates that possible shared mechanisms, in which the altered action of neurotrophic factors is involved, contribute to the pathogenesis of neurodegenerative and neuropsychiatric disorders, including both AD and major depressive disorder [5]. A recent study showed depression-like behaviors, cognitive impairment, and hippocampal BDNF downregulation in the offspring of mother mice that underwent sleep deprivation during pregnancy [6]. Thus far, a variety of clinical and experimental investigations suggest the possible contribution of the changed expression of neurotrophins in the pathophysiology of different neuropsychological disorders (see [7]).

Current studies also suggest that GDNF, IGF1, and bFGF, which are growth factors involved in various neuronal processes (including neuroprotection and the stimulation of intracellular signaling, which occurs in the BDNF/TrkB system), are associated with AD. In this review, we discuss current studies concerning the BDNF/TrkB system or other growth factors in AD, and new drug candidates affecting BDNF signaling for AD treatment.

## 2. Neurotrophins, Receptors, and Intracellular Signaling

Neurotrophins have multiple roles in the PNS and CNS. The neurotrophin family consists of NGF, BDNF, NT-4/5, and NT-3, and their contribution to cell maintenance and survival in the neuronal population has been extensively investigated. The specific high-affinity receptors for each neurotrophin, Trk receptors, are a family of tyrosine kinase receptors. TrkA is the high-affinity receptor for NGF; similarly, TrkB binds to both BDNF and NT-4, and TrkC is for NT-3, and the low-affinity common receptor for neurotrophins, p75NTR, also has functional roles in the PNS and CNS [1] (Figure 1). Among these neurotrophins and their receptors, the contribution of the BDNF/TrkB system to cell survival and synaptic function has been especially investigated [8]. The BDNF/TrkB system exerts a positive influence on neurons, including the potentiation of synaptic plasticity (the basis for learning and memory) and cell protection against intrinsic and extrinsic stress. The activation (phosphorylation) of TrkB, which is stimulated after binding with BDNF (or NT-4), triggers three signaling pathways, including the MAPK/ERK, PI3K/Akt, and PLCγ cascades. During development, in the establishment of accurate and adequate network connections of neurons with targets, limiting the quantity of neurotrophins derived from target tissues is critical since the regulation of the number of surviving neurons depends on each neurotrophin [9]. Generally, the promotion of cell survival is dependent on the PI3K/Akt pathway [8]. Growing evidence demonstrates that the MAPK/ERK signal pathway stimulates the activation of anti-apoptotic proteins to promote neuronal survival, and the PI3K/Akt pathway suppresses pro-apoptotic proteins [10]. Furthermore, MAPK/ERK signaling is also required for a variety of neuronal processes, including cell differentiation and synaptic maturation. We previously reported that the upregulation of synaptic proteins, which are essential for the release of neurotransmitters, was enhanced by BDNF in cultured cortical neurons [11]. It has been recognized that the PLCγ pathway plays an important role in intracellular Ca^2+^ increase by stimulating intracellular Ca^2+^ storage to regulate neurotransmitter release (see Figure 1).

Truncated forms also contribute to the physiological action of BDNF. Truncated TrkB.T1, TrkC.T1, and TrkB.Shc receptors were demonstrated to be biologically relevant splice variants (see [12]). The expression of TrkB.Shc is observed in the brain, while TrkB.T1 is distributed in the kidney, heart, and pancreas, in addition to the brain. Interestingly, it was reported that deficits in long-term potentiation (LTP)/depression (LTD) were observed following TrkB.T1 overexpression in animals, suggesting a functional contribution of the truncated form to synaptic plasticity [13]. Dorsey et al. (2012) showed enhanced nerve-evoked muscle tension and increased activation of full-length TrkB due to contractile activity in TrkB.T1-null mice [14], suggesting an inhibitory influence of truncated forms on the full-length TrkB form.

It is possible that functional interaction between p75NTR and TrkB affects neuronal functions, including cell survival. It is well known that neurotrophins are synthesized as proneurotrophins (lager-molecular-size precursors), and then, proneurotrophins undergo proteolytic cleavage to become mature forms which bind to Trks with high affinity [15]. Importantly, proneurotrophins, including proBDNF, preferentially bind to both p75NTR and sortilin (co-receptor), and trigger p75NTR-dependent apoptotic signaling [15]. For example, Volosin et al. (2006) reported that proNGF-induced cell death via p75NTR-mediated apoptotic signaling was not inhibited upon exposure to both mature neurotrophin and proneurotrophin in cultured basal forebrain (BF) neurons, which are known to be vulnerable to degeneration in AD [16]. proNGF-induced cell death was suppressed by infection with an adenovirus vector encoding activated ras to activate both the PI3K/Akt and MEK/ERK pathways, downstream of Trks [16]. Although the cytoplasmic domain of p75NTR has no catalytic activity [17], a variety of interactors associate with p75NTR and affect cell fate. p75 NTR is involved in various cell responses through interactions with neurotrophin receptor-interacting melanoma antigen homolog (NRAGE), neurotrophin receptor interacting factor (NRIF)1 and 2, p75NTR-associated cell death executor (NADE), tumor necrosis factor receptor-associated factor (TRAF) family proteins, and Ras homolog gene family member A (RhoA) [18,19,20,21,22]. As mentioned above, proneurotrophins bind to p75NTR with high affinity, and the proneurotrophins/p75NTR association causes cell death. In proNGF/p75NTR-induced cell death, an interaction of sortilin with p75NTR was important for death signal transduction [23] (Figure 1). The pro-domain of proBDNF was also associated with sortilin, and this association was involved in proBDNF/p75NTR-induced cell death [24,25].

## 3. Roles of BDNF/TrkB System in CNS Neurons

As mentioned above, the BDNF/TrkB system has multiple roles in developing and mature CNS neurons. In particular, the regulation of synaptic plasticity by BDNF has been intensively examined [26]. Luikart et al. (2005) found a decreased number of synapses after TrkB ablation in the hippocampal Schaffer collateral synapse [27]. Using cultured hippocampal neurons, Ji et al. (2010) found that the transient activation of TrkB was induced by an acutely increased concentration of BDNF, resulting in the promotion of neurite elongation and enlargement of the spine head [28]. In contrast, a gradually increased concentration of BDNF achieved sustained activation of TrkB, which was important for the branching of neurites and elongation of the spine neck [28]. It was also reported that BDNF increased the spine density of hippocampal neurons in the CA1 region by activating ERK signaling [29].

The expression of synapse-related molecules is also regulated by the BDNF/TrkB system. It is well recognized that vesicle-associated synaptic proteins (SV-proteins; synapsin I, synaptotagmin, synaptophysin, etc.), and plasma membrane-associated synaptic proteins (PM-proteins; syntaxin, SNAP25, etc.) are involved in the release of exocytotic neurotransmitters, including glutamate and gamma-aminobutyric acid (GABA) [30]. We previously reported that the BDNF-dependent upregulation of SV-proteins, such as synapsin I, synaptotagmin, and synaptophysin (not PM-proteins; syntaxin or SNAP25), occurred through the PLCγ and ERK pathways [31]. The expression levels of glutamate receptors are also regulated by BDNF. BDNF increased the hippocampal expression levels of alpha-amino-3-hydroxy-5-methyl-4-isoxazole propionic acid (AMPA) receptor subunits, including GluR1, GluR2, and GluR3 [32]. Furthermore, the mRNA levels of NMDA glutamate receptors, such as NR1, NR2A, and NR2B subunits, were also upregulated by BDNF [33]. We also found that BDNF upregulated these glutamate receptors (NR2A, NR2B, and GluR1) and presynaptic proteins (SNAP25, and synapsin I) in developing hippocampal neurons, and ERK signaling was required for synaptic upregulation [34].

Growing evidence has also demonstrated the involvement of BDNF in neurotransmitter release. Kang & Schuman (1995) found significant enhancement of synaptic transmission at the Schaffer collateral-CA1 synapses by exogenous BDNF and NT-3 (but not by NGF). In their study, they reported that the potentiation of synaptic strength by BDNF and NT-3 required Trk receptor activation [35]. Using cultured hippocampal neurons, it was demonstrated that both the frequency and amplitude of excitatory postsynaptic currents were potentiated by BDNF [36]. In cultured cortical neurons, we found that exogenous BDNF elicited the release of glutamate, which was dependent on the PLCγ-pathway [37]. The BDNF-induced glutamate release depended on the intracellular Ca^2+^ increase via activation of the PLCγ/IP_3_ system [37]. LTP, one of the most well-examined types of synaptic plasticity, is also induced through the BDNF/TrkB system. Using a virus-mediated approach, Lin et al. (2018) conducted the deletion of hippocampal BDNF in the CA1 or CA3 areas of the Schaffer collateral pathway, and revealed that the induction of LTP was dependent on presynaptic BDNF, although the maintenance of LTP required postsynaptic BDNF [38]. They also showed that LTP formation depended on postsynaptic TrkB and LTP maintenance, and relied on the presynaptic TrkB [38].

Since decreased neurogenesis is suggested to be involved in AD pathogenesis, the basic mechanism underlying the contribution of the BDNF/TrkB system to neurogenesis is an important issue. In order to examine the influence of BDNF in neurogenesis, Scharfman et al. (2005) performed two-week hippocampal infusion of BDNF with osmotic pumps implanted unilaterally into the dorsal hilus of rats [39]. One month after the BDNF infusion, BrdU(+)/NeuN(+) double-positive granule cells were increased [39]. Consistently, the knockdown of BDNF in the hippocampal dentate gyrus (DG) caused significant downregulation of neurogenesis [40].

Interestingly, growing evidence suggests a close connection between physical exercise and increased neurogenesis via the upregulation of BDNF [41]. In addition to BDNF and NGF, other factors, including vascular endothelial growth factor, are upregulated within the brain after exercise [5]. Importantly, the activation of neurogenesis upon being administered antidepressants or performing a wheel-running exercise did not occur when TrkB was knocked down [42], indicating the critical role of the BDNF/TrkB system in exercise-stimulated neurogenesis. Dong et al. (2022) showed the benefits of exercise in hippocampal synaptic plasticity [43]. Using female C57BL/6J mice, they found diminished performance in an object-location memory task after 7 days of sedentary delay following an initial 14 days of voluntary exercise (wheel running), which was restored by 2 days of reactivating the exercise after the sedentary delay. Enhanced hippocampal LTP in the CA1 region was also observed, which persisted throughout the sedentary delay and following the reactivation of exercise, although no difference was displayed in the mRNA expression of hippocampal BDNF between the exercise and control conditions [43]. It has been demonstrated that multifactor combinations, including physical exercise and visual and auditory stimulation, play a positive role in the treatment of AD [44]. Using a mouse model of AD created via an injection of Aβ oligomers into the bilateral DG regions, Li et al. (2022) found that hippocampal neurogenesis was increased by voluntary wheel exercise and involuntary treadmill running in combination with acousto-optic stimulation [44]. As expected, these multifactor combinations decreased the Aβ-induced impairment of learning and memory function, and upregulated the protein expression of postsynaptic density 95, synaptophysin, BDNF, and TrkB [44]; this suggests a relationship between physical exercise and neurogenesis via activation of the BDNF/TrkB system. The combined effect of resistance training (RT) and Urtica dioica extract (Ud, a traditional herb used as anti-inflammatory and antioxidant agent) on aging and memory impairment has been investigated [45]. Decreased hippocampal levels of NGF and BDNF and impaired cognitive function in aged rats were restored by regular RT using a ladder and an incline for five weeks (starting at 24 weeks old) with the administration of Ud [45]. Recently, researchers have also reported positive effects of endurance training (ET) and MitoQ (an antioxidant that accumulates in mitochondria) on memory and learning function [46]. Rats that were trained on a treadmill for 8 weeks and administered MitoQ through their drinking water showed higher performance in spatial memory (Morris water maze test) than those with ET or MitoQ alone. Furthermore, modulating parameters that are involved in hippocampal neurogenesis, including the upregulation of vascular endothelial growth factor (VEGF) and BDNF in the hippocampal tissue, were also confirmed [46]. Interestingly, the observation of beneficial effects of physical exercise on memory impairment in the offspring of old mice has been reported [47]. It was revealed that impaired spatial learning memory, the downregulation of hippocampal PSD95 and BDNF levels, decreased neurogenesis, and enhanced hippocampal apoptotic cell death were observed in the offspring of old mice; however, these impairments were reversed by running using a treadmill once daily for 6 days/week for 4 consecutive weeks [47]. Recently, using a stroke animal model, a positive influence of voluntary exercise on mesenchymal stromal cell (MSC) transplantation therapy was demonstrated. Yabuno et al. (2023) found that neurological deficits in rats inflicted with transient middle cerebral artery occlusion were significantly reversed after a combination of voluntary exercise (wheel running) and intracerebral transplantation of human modified bone marrow-derived MSCs [48]. Importantly, endogenous neurogenesis in the subventricular zone (SVZ) and DG, and mRNA levels of VEGF and BDNF, were also upregulated by mice undergoing the combination of exercise and MSC transplantation compared to those in single therapy or without therapy [48]. In addition, using an animal model of Parkinson’s disease (PD), which is also one of the most common neurodegenerative diseases, the upregulation of BDNF and an increase in neurogenesis following exercise was demonstrated. Leem et al. (2022) reported that a rotarod walking exercise decreased nigrostriatal degeneration in a mouse model of PD created via the administration of 1-methyl-4-phenyl-1,2,3,6-tetrahydropyridine (MPTP) [49]. In the animal model, they confirmed significant inhibition of reduced neurogenesis (SVZ, subgranular zone, substantia nigra, and striatum areas) caused by MPTP. Furthermore, it was revealed that the exercise induced the phosphorylation of adenosine monophosphate-activated protein kinase and the upregulation of BDNF. These findings suggest that physical exercise exerts a beneficial influence on neurogenesis and the BDNF/TrkB system, leading to neuroprotection and the positive regulation of neural function. To further understand the relationship between physical exercise and the regulation of neurogenesis, detailed information concerning the contribution of BDNF (pro or mature forms) and its receptors (TrkB or p75NTR) is needed. Furthermore, it is well known that the release of myokines (from skeletal muscles), adipokines (from adipose tissues), hepatokines (from the liver), and osteokines (from bone) occurs during physical exercise (see [50]). These exerkines circulate via the bloodstream and cross the blood–brain barrier, resulting in the regulation of neuronal function [50]. It would be interesting to further study possible interaction of the BDNF/TrkB system with exerkines in brain.

Neurogenesis has a pivotal role in the recovery process of the brain after stroke. Recently, Deshpande et al. (2022) demonstrated the critical contribution of p75NTR to the migration of neural stem cells, which were produced in the SVZ after stroke [51]. It was revealed that p75NTR KO mice exhibited reduced SVZ neural stem cell migration towards the lesion area after cortical brain injury [51].

## 4. BDNF/TrkB System and AD Models

Sen et al. (2015) reported the increased nuclear translocation of histone deacetylases, HDACs, by apolipoprotein E4 (ApoE4), one of the genetic risk factors for AD [52]. It was shown that the nuclear translocation of HDACs caused the downregulation of BDNF in human neurons, although ApoE3 induced upregulation of BDNF through the acetylation of histone 3. They also found that Aβ oligomers mimicked the action of ApoE4. Interestingly, ApoE3 indued protein kinase C ε (PKCε), and PKCε activation reversed the nuclear import of HDACs by ApoE4 and the Aβ oligomer, resulting in the inhibition of BDNF downregulation [52]. Using human neuroblastoma (SH-SY5Y) cells, it was demonstrated that significant downregulation of basal BDNF after Aβ treatment occurred through CREB transcriptional downregulation [53]. It is well recognized that the phosphorylated CREB protein is critical for the regulation of BDNF expression and its essential contribution in the CREB-mediated transcription system to the memory function process [54]. As evidence suggested that Aβ reduced BDNF expression mostly by decreasing the levels of pCREB protein, targeting of the CREB/BDNF pathway against Aβ toxicity was considered a beneficial approach to improving AD behaviors [54]. In vivo analysis revealed the downregulation of BDNF by Aβ. Xia et al. (2017) found that BDNF protein expression in both the cerebral cortex and the hippocampus of amyloid precursor protein/presenilin-1 (APP/PS1) Tg mice was markedly downregulated at the ages of 3 and 9 months [55]. Furthermore, they performed cerebroventricular injection (i.c.v) of Aβ1–42 oligomers into the C57BL/6 mice, and found reduced levels of BDNF protein in the cortical and hippocampal regions. Importantly, Aβ deposition and cognitive decline (determined using the Morris water maze test) in the APP/PS1 Tg mice were improved following BDNF treatment [55]. Compared with mice, rats are recognized to be genetically, physiologically, and morphologically closer to humans [56]. It has been reported that McGill-R-Thy1-APP transgenic rats, which overexpresses human APP751 by incorporating the Swedish and Indiana mutations with the Thy1.2 promoter [57], displayed differential dysregulation of BDNF or NGF expression [58]. As expected, progressive Aβ accumulation in the hippocampus and cerebral cortex in APP transgenic rats was observed. It was also revealed that mRNA levels of BDNF were significantly decreased in the early stages of amyloid pathology (before Aβ plaques appeared), although unchanged mRNA levels of NGF were confirmed, even at the advanced stages. Remarkably, significant upregulation of the NGF precursor (but not the BDNF precursor) was observed in APP transgenic rats [58], implying the involvement of a proNGF-mediated negative influence on AD-like phenotypes.

Higher expression of pTau (phosphorylated Tau) is considered to be a critical hallmark of AD. A recent study using postmortem brain tissues and fluid samples revealed lower levels of proBDNF in the frontal and entorhinal cortices in AD samples compared with healthy controls [59]. Furthermore, AD postmortem brains exhibited decreased density of hippocampal TrkB expression compared with the controls. Interestingly, higher serum proBDNF levels correlated with lower hippocampal proBDNF or higher pTau levels [59], suggesting a close relationship between BDNF (and/or TrkB) and pTau in the pathology of AD. Wang et al. (2019) reported that the application of the anti-BDNF antibody to inhibit the phosphorylation of TrkB triggered the activation of both Janus kinase 2/signal transducer and activator of transcription 3 (JAK2/STAT3) signaling in cultured rat neurons [60]. They found significant upregulation of inflammatory cytokines (IL-1β, IL-6, and TNFα) caused by downregulation of the BDNF/TrkB system, which activated the JAK2/STAT3 pathway, resulting in the elevation of CCAAT-enhancer binding proteinβ (C/EBPβ). Interestingly, the upregulation of C/EBPβ induced the expression of δ-secretase, leading to the cleavage of both APP and Tau, and promoting neuronal cell death [60].

All the above-mentioned studies suggest that decreased BDNF/TrkB signaling is strongly involved in the pathogenesis of AD, and the development of methods to increase BDNF in patients’ brains is expected to lead to treatment of the disease.

## 5. BDNF/TrkB System and Neuroprotection Drug Candidates for AD

Upregulation of the BDNF/TrkB system plays an important role in determining the protective effects of traditional medicine in neurodegenerative disease models. Using an experimental AD model, the effectiveness of Yuk-Gunja-Tang (YG), a Korean traditional medicine, was reported [61]. In scopolamine-induced memory impairment in C57BL/6 mice, the administration of YG improved impaired memory function in the Y-maze, passive avoidance, and novel object recognition tests. Interestingly, the administration of YG decreased cell death, and increased the expression of BDNF, in addition to levels of pERK and CREB [61].

As mentioned above, the hyperphosphorylation of Tau and its aggregation are typical neuropathological hallmarks of AD. Lin et al. (2022) examined the protective effects of analogous compounds of LM-031 (a coumarin derivative) on SH-SY5Y cells expressing ΔK280 tauRD-DsRed folding reporter [62]. ΔK280 tauRD is a deletion mutation of Tau found in patients with tauopathies [63,64]. Using theΔK280 tauRD-DsRed SH-SY5Y, the authors found that the LM-031 analogs LMDS-1 to -4 reduced Tau aggregation. They also found that LMDS-1 and LMDS-2 decreased the activity of caspase-1, caspase-6, and caspase-3, and the effect was reversed after the knockdown of TrkB, suggesting involvement of the BDNF/TrkB system in neuroprotection by LM-031 analogs [62]. Using SH-SY5Y cells expressing Aβ-GFP, the protective effects of the LM-031 analogs LMDS-1 to -4 have also been demonstrated. Chiu et al. (2022) discovered, via computational docking, that these LMDS compounds had the potential to bind to TrkB, and displayed anti-aggregation effect in SH-SY5Y cells expressing Aβ-GFP [65]. Although the overexpression of Aβ-GFP caused significant downregulation of pERK, pAkt, and pCREB, LMDS compounds restored the decreased levels of these phosphorylated molecules. The recovery effects of LMDS compounds were inhibited after the knockdown of TrkB [65].

Since both amyloid plaques, through the accumulation of Aβ, and neurofibrillary tangles (NFTs), through the accumulation of soluble Tau, are neuropathological hallmarks of AD, the possibility of a relationship between Tau toxicity and Aβ aggregation in the regulation of BDNF expression is very interesting. It was demonstrated, using in vivo system, that the overexpression of Tau caused downregulation of BDNF. Rosa et al. (2016) showed significant downregulation of BDNF mRNA in a transgenic mouse model of human Tau expression with or without NFTs [66]. Remarkably, they found that mice overexpressing Aβ also exhibited BDNF downregulation, although the BDNF expression was comparable to that in wild-type mice when crossed with Tau knockout mice, suggesting that Tau may be involved in BDNF downregulation caused by Aβ [66].

Recently, the therapeutic potential of exosomes prepared from bone marrow mesenchymal stem cells (BMSC-exos) was demonstrated. Liu et al. (2022) performed the administration of BMSC-exos via lateral ventricle or caudal vein injection into sporadic AD (SAD) mouse models, produced via the intracerebroventricular injection of streptozotocin, and found that lateral ventricle (but not caudal vein) administration decreased Aβ1–42, pTau, IL-1β, IL-6, and TNF-α, and upregulated BDNF protein [67]. Importantly, AD-like behaviors (assessed using an open field test and a novel object recognition test) in the disease model mice were improved through BMSC-exos administration [67].

Using the SAD mouse model established via streptozotocin injection, the neuroprotective potential of bergapten (BG, furanocoumarin found in a variety of medicinal plants) and tadalafil (TAD, phosphodiesterase 5-inhibitor) was examined [68]. In addition to AD-like pathologies, including Aβ deposition and Tau aggregation, impaired insulin and Wnt/β-catenin signaling, neuroinflammation, and dysfunction of the autophagy system were observed in the SAD mice. Importantly, impaired cognitive function (assessed using the Morris water maze test and the object recognition test) in the SAD animals was improved through the administration of BG or TAD. The administration of BG or TAD also reduced Aβ expression and Tau hyperphosphorylation, and increased the hippocampal expression of pAkt, phosphorylated glycogen synthase kinase (pGSK)-3β, and BDNF immunoreactivity [68].

To improve behavior in neurodegenerative disorders, including AD, natural products that upregulate the BDNF/TrkB system have been the focused of research [69] (see Figure 2). In particular, the beneficial action of flavonoids for achieving neuroprotection in the pathogenesis of AD has been intensively investigated [70]. Importantly, 7,8-dihydroxyflavone (7,8-DHF) acted as a TrkB agonist by binding to the extracellular domain of TrkB, and exerted BDNF-like activity [71]. Synthetic prodrug R13 has been produced to improve the poor oral bioavailability of parent compound 7,8-DHF [72]. The R13 showed significant upregulation of 7,8-DHF’s pharmacokinetic profile. It was also confirmed that the chronic oral administration of R13 induced TrkB activation and inhibited Aβ deposition, hippocampal synapse loss, and decreased memory function in 5XFAD mice [72]. Furthermore, using the 5XFAD mice, Li et al. (2022) reported hippocampal activations of TrkB, ERK, and Akt signaling after intragastrical treatment with R13 [73]. Because mitochondrial dysfunction is related to AD [74], they performed mitochondriomics analysis, and found that R13 increased the levels of ATP and the expression of mitochondrial oxidative phosphorylation-related proteins, such as complex I, II, III, and IV, in the hippocampus of AD model mice. The downregulation of Aβ plaque and pTau following R13 treatment was also confirmed in their systems. Interestingly, it has been reported that chrysin (5,7-dihydroxyflavone) potentially binds to TrkA, TrkB, and p75NTR [75]. Hypothyroidism (which causes AD-like cognitive function) model mice, induced through continuous exposure (31 days) to methimazole in drinking water, exhibited memory deficits (Morris water maze test) and downregulation of BDNF and NGF. However, improved spatial memory function and increased BDNF expression in the hippocampus, and increased NGF in both the hippocampus and prefrontal cortex, were observed after the intragastrical administration of chrysin for 28 consecutive days [75]. Using an AD mouse model induced through an injection (i.c.v) of Aβ1–42, the effect of Kaempferide (KF), one of the flavonoids derived from *Alpinae oxyphylla* Miq, was examined. After Aβ1–42 exposure, KF administration (i.c.v) for five consecutive days was performed. As expected, behavioral tests (Y-maze test and Morris water maze test) revealed that KF prevented cognitive decline in the mice that received Aβ1–42 injection. The administration of KF upregulated the activities of BDNF/TrkB and CREB signaling in the hippocampus [76]. It has also been demonstrated that the amyloidogenic pathway is inhibited by a plant-derived flavonoid compound. Yin et al. (2018) investigated the effect of icariside II (ICS II) on cognitive deficits and Aβ levels in chronic cerebral hypoperfusion (CCH) rats [77]. Significant cognitive deficits and increased hippocampal Aβ1–40 and Aβ1–42 levels were observed in the CCH rats, induced via bilateral common carotid artery occlusion. the oral administration of ICS II (28 days) abolished cognitive deficits. Furthermore, the expression levels of APP and β-site amyloid precursor protein cleavage enzyme 1 (BACE1) were decreased following ICS II treatment. On the other hand, increased levels of a disintegrin and metalloproteinase domain 10 (ADAM10), and insulin-degrading enzyme (IDE), were observed following ICS II treatment. ICS II also enhanced the expression of BDNF and TrkB, and levels of pAkt and pCREB, suggesting that inhibition of the amyloidogenic pathway via BDNF/TrkB/CREB signaling was involved in the beneficial action of ICS II [77].

Growing evidence indicates downregulation of the BDNF/TrkB system in AD postmortem brains and in vitro and in vivo AD models. Neuroprotection, such as improved neuronal function and cell death inhibition, through potentiation of the BDNF/TrkB system has been demonstrated to contribute to the effectiveness of drug candidates, including flavonoids.

## 6. Other Neurotrophic Factors in AD

### 6.1. Glial Cell Line-Derived Neurotrophic Factor (GDNF)

GDNF, one of the most widely known neurotrophic factors, was discovered as the first member of the GDNF family of ligands in the CNS in a rat glial cell line [78]. Generally, it is well known that glial cells, including astrocytes, produce GDNF [79]. GDNF is also produced by dopaminergic neurons in the substantia nigra and has protective effects on dopaminergic and other types of neurons. GDNF and its receptors are widely distributed throughout the brain and are present in the spinal cord, kidneys, and other organs [80].

Regarding the role of GDNF in the CNS, studies using heterozygous knockout animals have shown that GDNF is involved in neurogenesis and the regulation of brain functions, such as the modification of mental conditions and suppression of drug dependence [81,82,83].

GDNF family receptor alpha (GFRα) is bound to the cell membrane by a glycosylphosphatidylinositol (GPI) anchor and acts as the main receptor for GDNF [84]. Four types of GFRα, including GFRα1, α2, α3, and α4, are known. When GFRα1 binds to the GDNF homodimer, the receptor interacts with rearranged during transfection (RET), and activates it. Activated RET triggers multiple signaling pathways, including the PI3K/Akt and MAPK/ERK signaling pathways [85]. While RET alone has pro-apoptotic activity [86], the activation of these downstream signaling pathways is thought to be responsible for GDNF-induced neuronal survival, as both the PI3K/Akt and MAPK/ERK signaling pathways prevent neuronal cell death [87]. Interestingly, the existence of GDNF/GFRα signaling without RET has also been demonstrated. In this case, the GDNF/GFRα system promotes the activation of c-fos gene transcription and the survival of neurons (that do not express RET) through the activation of Src family kinases and the pLCγ pathway [88].

The pathological role of GDNF in AD is not completely understood; however, several studies have shown that depletion of this neurotrophic factor appears to be associated with symptoms and pathologies of AD, and could be a potential therapeutic option for neurodegeneration (Table 1). Ghribi et al. (2001) demonstrated that the administration of GDNF exerted significant neuroprotection in AD model rabbits against aluminum-induced apoptosis by upregulating bcl-XL and abolishing caspase-3 activity [89]. In addition, it was reported that GDNF protected both neurons and glial cells from kainate-induced excitotoxicity and oxidative stress [90].

Marksteiner et al. (2011) reported that GDNF was reduced in the plasma and increased in the cerebrospinal fluid of patients with early-stage AD [91]. Straten et al. (2011) showed similar results of reduced serum GDNF and increased in the cerebrospinal fluid of patients with AD [92]. Another study also reported the downregulated expression of BDNF, NGF, and GDNF in patients with mild cognitive impairment and moderate AD [93]. Moreover, it has been suggested that the downregulation of GDNF causes excessive glutamate release by dysregulating GLT-1, a glutamate transporter that is responsible for removing glutamate from the synaptic cleft [94], resulting in the excitotoxicity that precedes dopaminergic degeneration [95]. It has been demonstrated that dopaminergic degeneration is strongly associated with AD, and pharmacological approaches to enhancing the dopaminergic system improve synaptic dysfunction and memory impairment in patients [96]. Interestingly, the upregulation of endogenous GDNF in GDNF hypermorphic mice (Gdnfwt/hyper) suppressed a gradual decline in cholinergic transmission, which is one of the hallmarks of dementia [97]. Airavaara et al. (2011) identified novel GDNF isoforms and found that the mature GDNF peptide was significantly downregulated in the postmortem brains of AD patients [98].

As expected, a previous study reported that GDNF was reduced in 3xTgAD mice (a transgenic AD mouse model) although 6 months of voluntary exercise reversed this downregulation [99]. Other studies suggest that gene therapy is a safe and promising treatment for AD. Revilla et al. (2014) used recombinant lentiviral vectors to overexpress the GDNF gene in 3xTgAD mouse hippocampal astrocytes in vivo. After overexpressing GDNF, 10-month-old 3xTgAD mice displayed preserved learning and memory functions [100]. These findings suggest that GDNF overexpression is effective at preventing cognitive loss and memory impairment in AD; therefore, it is necessary to uncover the mechanisms of GDNF-dependent neuroprotection and its role in the crosstalk between glia and neuronal cells.

### 6.2. Basic Fibroblast Growth Factor (bFGF, or FGF-2)

bFGF, also known as FGF-2, is a heparin-binding protein with a molecular weight of 18–34 kDa. This growth factor was identified in 1974 in the bovine pituitary gland as a molecule that promoted fibroblast proliferation [101]. bFGF, initially named for its role in fibroblast proliferation, is a multifunctional growth factor involved in a variety of cellular processes. During development, bFGF contributes to mesoderm induction, anterior–posterior axis patterning, limb formation, and neurogenesis [102]. In adulthood, bFGF contributes to angiogenesis and wound healing by acting on various cell types [103].

bFGF is primarily produced as a polypeptide of 155 amino acids, resulting in an 18 kDa protein. Isoforms of 22 kDa (196 amino acids), 22.5 kDa (201 amino acids), 24 kDa (210 amino acids), and 34 kDa (288 amino acids) are also synthesized due to alternative start codons with N-terminal extensions of 41, 46, 55, or 133 amino acids, respectively [104,105]. Typically, the low-molecular-weight (LMW) form of 18 kDa (155 amino acids) is located in the cytoplasm and secreted from cells, whereas the high-molecular-weight (HMW) form is delivered to the cell nucleus [106]. bFGF binds to FGF receptors (FGFRs) on the cell membrane. The binding induces the formation of FGFR dimers, and the dimerization triggers the activation (phosphorylation) of tyrosine kinase domains, which act as docking sites for other signaling proteins. Heparan sulfate is involved in the dimerization process [107]. The signaling pathways stimulated through the activation of FGFR are the Ras, PLC-γ, Janus kinase (JAK) signal transducer and activator of transcription (Stat), and PI3K/Akt pathways, which regulate a variety of gene expression and cellular processes [108]. In the CNS, bFGF is expressed in various cell populations, such as neurons, astrocytes, oligodendrocytes, and microglia [109,110]. It was shown that bFGF attenuated glutamate-triggered cell death by preventing intracellular Ca^2+^ increase and suppressing oxidative radical accumulation and mitochondrial dysfunction, and inhibited neurodegeneration mediated by Aβ [109,111,112,113,114,115,116,117]. Interestingly, bFGF is considered to have therapeutic potential as it increased the number of cholinergic neurons and improved their survival in a rat unilateral fimbria–fornix model of acute cholinergic neuronal degeneration [118]. It has been suggested that FGFR1-increased CD200 (a membrane glycoprotein that controls microglial activity and promotes neuronal survival) is involved in these protective effects exerted by bFGF [119].

A recent report showed that both the LMW and HMW isoforms of bFGF protected cortical astrocytes against Aβ1–42-induced toxicity by activating the PI3K/Akt signaling pathway [105]. Another study demonstrated that the HMW isoforms of bFGF had stronger neuroprotective effects than LMW isoforms in rat hippocampal neurons [120]. Another study showed that the HMW isoform was significantly downregulated in AD patients, although no difference in their levels of LMW isoforms was found compared with age-matched healthy controls [121].

bFGF can cross the blood–brain barrier (BBB) when applied systemically. Using APP23 transgenic mice as a model of amyloid pathology in AD, Katsouri et al. (2015) found that exogenous bFGF administration improved Tau pathology and spatial memory impairment by downregulating BACE1 [121]. Moreover, using a mouse model of AD (APP+PS1), Kiyota et al. (2011) reported that bFGF gene delivery to the hippocampus significantly reduced Aβ synthesis, facilitated its clearance, and restored spatial learning activity (usinhg a radial arm water maze test) [122]. Intranasal administration appeared to be more effective than intravenous injection for delivering bFGF into the brain. Evidence demonstrates that the intranasal administration of bFGF nanoparticles promotes Aβ clearance, improves LTP and spatial learning, and stimulates neurogenesis in the SVZ in AD animals [123,124,125,126].

### 6.3. Insulin-like Growth Factor 1 (IGF-1)

IGF-1 is a pleiotropic hormone whose structure is similar to that of insulin. IGF-1 is produced in various tissues, including the brain, is recognized to be an essential factor for cell growth and differentiation, and contributes to the maintenance of various organs [127,128]. IGF-1 binds to the IGF-1 receptor (IGF1R) and/or insulin receptor (IR). The activation of receptor tyrosine kinases results in the phosphorylation of insulin receptor substrates (IRSs), which are linked to IGF1R and IR. Adaptor proteins bind to the phosphorylated IRSs, thereby transmitting signals to downstream pathways, including PI3K/Akt [129]. IRSs have pleckstrin homology and phosphotyrosine binding (PTB) domains at the N-terminus, and PI3K-, Grb2-, and SH2-binding domains at the C-terminus. Mammals have four types of IRS (IRS1-4) [130]. IRS1 and IRS2 are ubiquitously expressed in almost all organs of the body. IRS3 is observed only in adipocytes (human IRS3 is a pseudogene). IRS4 is mainly expressed in the hypothalamus, showing different expression patterns from those of IRS1 and IRS2 [131,132]. Although these IRSs display differences in their structure and expression patterns, common downstream signaling pathways are activated. Unlike the peripheral IGF-1-target tissues, there are significant differences in the distribution and expression between IR and IGF1R in the brain, although their expression in some brain regions is still unclear [130]. Brain-specific IR-deficient (bIR−/−) and neuron-specific IGF1R-deficient (nIGF1R−/−) mice have been generated [133]. To examine the possible involvement of IR and/or IGF1R in the pathogenesis of AD, the AD model mice (Tg2576) were crossed with their respective knockout mice. Tg2576 mice crossed with nIGF1R−/− (Tg2576 X nIGF1R−/−) showed prolonged survival and improved AD pathology and cognitive dysfunction, similar to those crossed with IGF1R heterozygous (IGF1R+/−) mice (Tg2576 X IGF1R+/−) [134,135,136,137]. Another study demonstrated that heterozygous brain-specific IGF1R-deficient (bIGF1R+/−) mice showed improved mortality without affecting the pathology of Tg2576 mice (Tg2576 X bIGF1R+/−) [138]. Recently, Sohrabi et al. (2020) reported that the administration of IGF1R inhibitor ameliorated Aβ accumulation and neuroinflammation in transgenic AD model mice (AβPP/PS1) [139]. In contrast, bIR−/− mice exhibited anxiety- and depression-like behavioral abnormalities due to the dysfunction of mitochondrial biogenesis in the dorsal striatum and dopamine neurons [140]. After being crossed with Tg2576 mice (Tg2576 X bIR−/−), beneficial effects on the phenotype of AD were not observed [134].

Among the four IRSs (IRS1-4), IRS2 has been most thoroughly investigated. For example, it was reported that IRS2 KO mice exhibited a striking phenotype of juvenile lethality associated with insulin resistance and the development of severe diabetes [141]. Although IRS2 is expressed throughout the body, it is also widely distributed in the brain and highly expressed in neurons. Using double mutant mice generated by crossing IRS2−/− mice with Tg2576 mice, it was revealed that IRS2 deletion improved all pathological features (decreased cognitive function and mortality in Tg2576 mice) [142]. Tg2576 mice crossed with nIGF1R−/− (Tg2576 X nIGF1R−/−) showed similar improvement [134]. These AD mouse model studies suggest that IGF-1 signaling exacerbates AD.

Several reports have demonstrated protective roles of IGF-1 in AD. The administration of growth hormone-releasing hormone (GHRH), an inducer of IGF-1, has been clinically confirmed to ameliorate cognitive impairment by increasing levels of N-acetylaspartyl-glutamate (NAAG) and GABA in AD patients [143,144]. Furthermore, IGF-1 exhibited protective action against Aβ25–35 toxicity through the PI3K/Akt pathway, and by inhibiting pro-apoptotic proteins such as p53 upregulated modulator of apoptosis (PUMA) and Bcl-2-associated X protein (Bax), in a human neuroblastoma cell line [145]. Carro et al. (2002) reported a strong correlation between brain Aβ levels and serum IGF-1 concentrations in rats. Interestingly, the prolonged administration of IGF-1 to aged rats achieved downregulation of brain Aβ levels to levels comparable with those of young rats [146]. This effect of IGF-1 on Aβ clearance was mediated by enhancing the transport of Aβ carrier proteins (transthyretin and albumin) into the brain by increasing the permeability of the choroid plexus, which forms the blood–cerebrospinal fluid (B-CSF) barrier [146]. Importantly, the disruption of IGF-1 signaling caused Tau phosphorylation in IRS2 KO mice (IRS2−/−) [147]. In addition, IGF-1 is recognized to play a role in neuroprotection and the promotion of neurogenesis [148,149,150]. These collective findings indicate that IGF-1 signaling could be a potential therapeutic target in AD (Table 1). Human and animal models demonstrate dichotomous effects of IGF-1 on the pathogenesis of AD, with both exacerbating and protective activity; therefore, further investigations are needed to better understand the multiple roles of IGF-1 and its signaling in the context of AD.

**Table 1 life-13-00647-t001:** Neurotrophic effects of GDNF, bFGF, and IGF-1 on AD models.

Target Factors	Experimental AD Models	Treatments	Neurotrophic Effects	References
GDNF	Intracisternal injection of aluminum complexes into rabbit brain	Intracisternal injection of 100 μL of 500 ng/mL GDNF	Inhibition of neuronal death by upregulating bcl-XL and abolishing caspase-3 activity	[89]
	3xTgAD mouse	6 months of GDNF overexpression using recombinant lentiviral vectors in hippocampal astrocytes	Preservation of learning and memory Upregulation of BDNF	[100]
bFGF	Rat primary cortical neurons treated with glutamate (10 μM, 30 μM) and Aβ25–35 (1.0 μM)	Application of bFGF (0.3, 1, and 3 ng/mL) to the culture at 24 h before Aβ25–35 and glutamate stimulation	Decreased neuronal damage elicited by Aβ25–35 and glutamate	[111]
	Rat hippocampal neurons treated with Aβ25–35 (20 μM)	Pre-treatment for 16 h with bFGF (10 ng/mL)	Suppression of reactive oxygen species (ROS) accumulation	[112]
	Rat primary hippocampal neurons treated with 20 μM Aβ1–42	Treatment with LMW and HMW bFGF (10 μg/mL, 50 μg/mL, and 200 μg/mL) for 24 h	Protection against Aβ1–42-induced neurotoxicity Activation of the ERK and Akt signaling pathways	[120]
	APP23 mouse	Subcutaneous injection of recombinant bFGF (20 μg/kg per day) for 21 days	Attenuation of spatial memory deficits and Aβ and tau pathologies by downregulating BACE1	[121]
	APP+PS1 mouse	Hippocampal AAV2/1(AAV serotype 2/1 hybrid virus)-bFGF injection	Reduced Aβ synthesis, and restored spatial learning in the radial arm water maze test	[122]
	APP+PS1 mouse	Intranasal administration of bFGF (0.5 μg/g)	Improvement of spatial learning ability, hippocampal LTP, and Aβ species accumulation in hippocampus	[126]
IGF-1	SH-SY5Y cells treated with Aβ25–35 (25 μM) or Aβ1–42 (1.5 μM) for 24 h	Application of IGF-1 (100 ng/mL) to the culture	Prevention of Aβ-induced cell death through activation of the PI3K/Akt pathway	[145]
	Aging rats (>18-mo-old)	Chronic administration of IGF-1 using minipumps (50 μg/kg)	Downregulation of brain Aβ levels by enhancing transport of Aβ carrier proteins	[146]

## 7. Conclusions and Future Perspectives

In developed countries, the increased number of patients with AD has been a major problem. With the advent of superior diagnostic techniques such as imaging tests and cerebrospinal fluid biomarkers, the accuracy of AD diagnosis in clinical settings has greatly improved. The outcomes of basic research using animal models have also deepened our understanding of the pathomechanisms of dementia. However, the development of drugs to treat the cause of AD has been delayed. Three cholinesterase inhibitors (donepezil, galantamine, and rivastigmine) and an NMDA receptor antagonist (memantine) are prescribed to patients with AD in clinical practice. These anti-dementia drugs are for symptomatic treatment, and while they may temporarily improve symptoms such as cognitive dysfunction, they cannot halt the progression of pathological changes in AD. Patients return to their pre-dose state within one to two years of taking the drug, and dementia becomes more severe in the long term. Therefore, the development of disease-modifying drugs that act directly on the pathophysiology of AD and have the effect of suppressing the progression of brain dysfunction is eagerly awaited.

Neurotrophic factors contribute to the growth, survival, and function of brain neurons, and have attracted attention for their application in the treatment of AD. In particular, BDNF, one of the most well-known neurotrophic factors, exhibits diverse physiological activities in the CNS, such as the maintenance of neuronal survival, the morphogenesis of neurites, and the regulation of synaptic plasticity, through its tyrosine kinase-type receptor, TrkB. When administered to the brains of AD animal models, BDNF inhibits the loss of nerve cells and restores neural functions, and thus, is expected to be used as a therapeutic agent for the disease. Moreover, accumulating evidence suggests that other neurotrophic factors, including NGF, GDNF, and bFGF, also have neuroprotective potential and are promising targets for AD.

Importantly, there are several obstacles that need to be resolved in clinical application of neurotrophic factors. For instance, neurotrophic factors are high-molecular-weight proteins and cannot cross the BBB when administered peripherally, making it difficult to use them as drugs for the treatment of AD. It is necessary to develop a drug delivery system that can deliver them to specific areas in the brain. Small-molecule compounds that can be transferred via the BBB and promote the synthesis of endogenous neurotrophic factors in the brain are an attractive and promising strategy. In fact, a lot of small compounds and natural products are reported to ameliorate AD pathology in animal models via upregulation of the BDNF/TrkB system. In some cases, however, drug candidates that had shown apparent beneficial effects on AD model animals failed to delay AD progression in human clinical trials. This is thought to be due to differences in species (e.g., human vs. mouse) and artificial phenotypes caused by the overexpression of causative genes. In this regard, induced pluripotent stem cell (iPSC) technology enables us to utilize AD patients’ derived human neurons or cerebral organoids for the development of more effective drugs with fewer adverse reactions. Other limitations regarding the clinical application of neurotrophic factors, such as a short half-life, difficulty in injecting appropriate doses, and side effects on organs and tissues (for example, due to TrkB or p75NTR), need to be resolved in the future. Nevertheless, harnessing neurotrophic factors will provide new preventive and therapeutic alternatives for AD and other neuropsychiatric disorders.

## Figures and Tables

**Figure 1 life-13-00647-f001:**
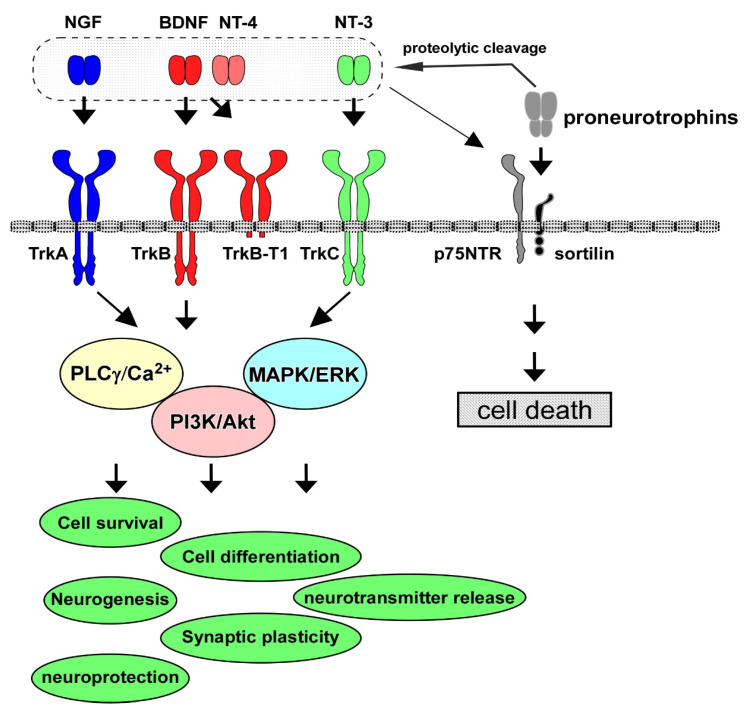
Neurotrophins and receptors in CNS neurons.

**Figure 2 life-13-00647-f002:**
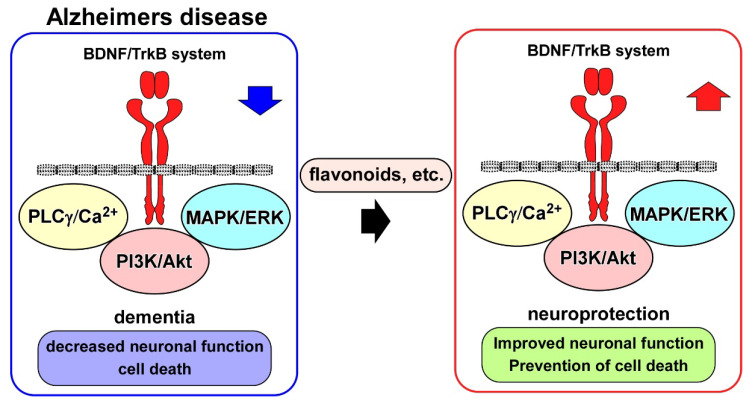
Downregulation of BDNF/TrkB system in Alzheimer’s disease (AD), and its upregulation by drug candidates. Blue and red arrows indicate downregulation and upregulation, respectively.

## Data Availability

No new data were created or reported in this review article.

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
