# Peer review of "Neurotrophins and Other Growth Factors in the Pathogenesis of Alzheimer’s Disease"

_life, 2023, doi:10.3390/life13030647_

Round 1
Reviewer 1 Report
I reviewed the manuscript "Neurotrophins and other growth factors in the pathogenesis of Alzheimer's disease" by Numakawa and Kajihara.
Overall, the text is well written and easy to follow; however, this manuscript looks like a book chapter and not a journal review. Most of the references are old, before 2015 or even before 2000. There is already plenty of reviews regarding the role of BDNF in AD in the literature (e.g. 10.14336/AD.2015.0825, published in 2016). Authors should focus on recent articles (after 2020) in order to publish a review on a scientific journal. The only chapter that is valuable for publication is Chapter 5, where authors report recent studies that bring novel information to the field.
Unfortunately, I must reject this manuscript.
Reviewer 2 Report
The reveiw was a resonable summation of what is know about neurotrophin function and its possible involvement in neurodegeneration. Table 1 was of interest and a useful compilation of what is known on Neurotrophic effects of GDNF, bFGF and IGF-1 on AD models.
Reviewer 3 Report
The review written by Numakawa and Kajihara highligths the neurotrophic factors molecular mechanims on AD onset and theyr action on the manteining of neuronal functions.Moreover, they explain the importance of them as drug targets in clinical approach.
It resuled be well written and clear.
Reviewer 4 Report
The authors reviewed the potential role of down regulation of neurotrophic factors, including BDNF and others, in the in the pathogenesis of Alzheimer's disease and the beneficial effects of upregulating neurotrophic factors, including by many natural compounds, in improving the pathology of Alzheimer's disease. Current drugs prescribed for AD, including cholinesterase inhibitors and NMDA receptor antagonist, only temporarily improve symptoms but don’t prevent the pathological progression in AD. Neurotrophic factors have the potential to slow down the pathological progression in AD by their neuroprotective effects.
The manuscript is informative and well organized, but needs some English editing.
Reviewer 5 Report
In this review, the authors systematize data on the expression/function of neurotrophic factors in the pathogenesis of neurodegenerative diseases, including Alzheimer's disease. Involvement of neurotrophic factors, in particular brain-derived neurotrophic factor (BDNF) and its high-affinity receptor TrkB, which perform several functions in the central nervous system, in particular neuronal maintenance, synaptic plasticity, and neurogenesis, which are closely related to the function of learning and memory. The article provides a detailed analysis of the involvement of BDNF / TrkB, as well as other neurotrophins in the pathogenesis of AD, with the search or screening of new drug candidates for the treatment of age-related dementia. In general, the review work makes a good impression with the depth of study and the high quality of the systematic analysis of literature data on the issue under study and deserves high praise. In the process of reviewing this article, the following minor comments and questions to the authors arose:
1. The first reference to the literature is still in print and has no confirmed reference. You need to add the necessary data for correct citation.
2. Very interesting and relevant are the data that depression is considered a risk factor for the development of AD, and that the BDNF/TrkB system, which plays a key role in maintaining neurons, including neuronal survival and synaptic functions, is suppressed in patients with both depression and AD. The last provisions must be supported by correct references and it is possible to develop this proposition in more detail in the Introduction section.
3. There is great interest in the data indicating a close relationship between physical exercise and increased neurogenesis through the activation of BDNF [38]. I believe that this fact needs more disclosure and discussion.
4. At the end of Section 4 "BDNF/TrkB System and AD Models", a conclusion should be made summarizing the presented and analyzed literature data. In the current version, this section contains only facts related to the BDNF/TrkB system for the AD model. It is necessary to analyze these facts and summarize the conclusion in the form of an imperative statement and / or hypothesis, which will significantly improve the perception of this section of the review.
5. At the end of Section 5 «BDNF/TrkB system and candidates for neuroprotection drugs for treatment of AD» need to be better structured (as done in section 6) and possibly subdivided for better readership.
After making appropriate changes to the text of the review article, the work can be published in the journal Life
Reviewer 6 Report
In the review entitled “Neurotrophins and other growth factors in the pathogenesis of Alzheimer's disease” authors describe the more significant results obtained from studies performed in the field of neurorophin effects in Alzheimer Disease (AD) pathology, in particular BDNF. Other neutrotrophins such as GDNF, IGF1, and bFGF have been also considered in the review. Results from both, in vivo and in vitro disease models as well as the few results from clinical trials performed in this context have been included in the review as well.
The review is important in the field of neurodegenerative diseases, in particular AD, for which there are not an effective therapy yet.
The manuscript is well organized and written in a fashioned English style and scientifically sound.
I have only very few comments that could help to improve the manuscript. They are listed below:
- Pg 2, line 73, “…since a regulation of the number…”, change with “since the regulation of the number…”
- Pg 4, lines 154-155, “Growing evidence has also demonstrated that an involvement of BDNF in the neurotransmitter release”, change better with “Growing evidence has also demonstrated the involvement of BDNF in the neurotransmitter release”
- Pg 4, lines 170-172, “…basic mechanism underlying contribution of BDNF/TrkB system to neurogenesis is important issue”, change with “…basic mechanism underlying the contribution of BDNF/TrkB system to neurogenesis is an important issue”.
- Pg 9, lines 370-371, “… reported that GDNF was reduced in the plasma of and increased in the cerebrospinal fluid of patients with early-stage AD”, change with “…reported that GDNF was reduced in the plasma and increased in the cerebrospinal fluid of patients with early-stage AD”.
- Pg 9, lines 372-373, “…reduced serum GDNF and increased one in the cerebrospinal fluid of patients with AD”, change with “…reduced serum GDNF and increased in the cerebrospinal fluid of patients with AD”.
- Pg 16. Lines 525-526, “When administered to the brains of animal models of AD”, change with “When administered to the brain of AD animal models”.
- Table 1 “Neurotrophic effects of GDNF, bFGF and IGF-1 on AD models”, would be much improved including lines to separate different information contained in it.
Round 2
Reviewer 1 Report
I appreciate the work that authors did to improve their manuscript. I still think that authors should focus even more on recent publications or novel perspectives to have a review article published in a journal, 5 new references are not enough to change my previous comment.
